# Electrodiagnostic Testing and Nerve Ultrasound of the Carpal Tunnel in Patients with Type 2 Diabetes

**DOI:** 10.3390/jcm11123374

**Published:** 2022-06-13

**Authors:** Bianka Heiling, Leonie I. E. E. Wiedfeld, Nicolle Müller, Niklas J. Kobler, Alexander Grimm, Christof Kloos, Hubertus Axer

**Affiliations:** 1Department of Neurology, Jena University Hospital, Friedrich Schiller University, 07747 Jena, Germany; leonie.wiedfeld@med.uni-jena.de (L.I.E.E.W.); niklasjohannes.kobler@med.uni-jena.de (N.J.K.); hubertus.axer@med.uni-jena.de (H.A.); 2Clinician Scientist Program OrganAge, Jena University Hospital, 07747 Jena, Germany; 3Department of Internal Medicine III, Jena University Hospital, Friedrich Schiller University, 07747 Jena, Germany; nicolle.mueller@med.uni-jena.de (N.M.); christof.kloos@med.uni-jena.de (C.K.); 4Department of Neurology, Tuebingen University Hospital, 72076 Tuebingen, Germany; alexander.grimm@med.uni-tuebingen.de

**Keywords:** carpal tunnel syndrome, diabetes mellitus, nerve conduction study, peripheral nerve ultrasound

## Abstract

In diabetic patients, controversies still exist about the validity of electrodiagnostic and nerve ultrasound diagnosis for carpal tunnel syndrome (CTS). We analyzed 69 patients with type 2 diabetes. Nerve conduction studies and peripheral nerve ultrasound of the median nerve over the carpal tunnel were performed. CTS symptoms were assessed using the Boston Carpal Tunnel Questionnaire. Polyneuropathy was assessed using the Neuropathy Symptom Score and the Neuropathy Disability Score. Although 19 patients reported predominantly mild CTS symptoms, 37 patients met the electrophysiological diagnosis criteria for CTS, and six patients were classified as severe or extremely severe. The sonographic cross-sectional area (CSA) of the median nerve at the wrist was larger than 12 mm^2^ in 45 patients (65.2%), and the wrist-to-forearm-ratio was larger than 1.4 in 61 patients (88.4%). Receiver operating characteristic analysis showed that neither the distal motor latency, the median nerve CSA, nor the wrist-to-forearm-ratio could distinguish between patients with and without CTS symptoms. Diagnosis of CTS in diabetic patients should primarily be based upon typical clinical symptoms and signs. Results of electrodiagnostic testing and nerve ultrasound have to be interpreted with caution and additional factors have to be considered especially polyneuropathy, but also body mass index and hyperglycemia.

## 1. Introduction

Carpal tunnel syndrome (CTS) is the most common entrapment neuropathy in the general population [1]. Risk factors for CTS are diabetes mellitus, obesity, metabolic syndrome, thyroid dysfunction, rheumatic diseases, and others [2,3]. It has been shown that the incidence of CTS is increased in diabetic patients [4]. Typical symptoms are numbness, predominantly nocturnal par- and dysesthesias, and/or neuropathic pain, which are associated with localized compression of the median nerve at the wrist, and weakness and atrophy of the thenar muscle later in the course of the disease [5,6]. 

Nerve conduction studies are the technical gold standard for diagnosis [7]. CTS typically shows an elongation of the distal motor latency (DML) and a decrease in sensory nerve conduction velocity (CV) measured over the wrist. Principally, sensory changes occur before motor changes and changes in latencies and CV precedes changes in the amplitudes of compound motor action potentials (CMAP) and sensory nerve action potentials (SNAP). It is recommended that nerve conduction studies for the diagnosis of CTS be performed in patients with clinical manifestations of CTS [8].

Recently, peripheral nerve ultrasound was revealed to give valuable additional information; so, it has been advised to perform nerve ultrasound in addition to electrodiagnostic testing [9]. Sonographic diagnosis is based on a swelling of the median nerve at the inlet of the carpal tunnel (at the level of the pisiform bone), where an increase in the cross-sectional area (CSA) of the median nerve can be measured [10,11]. In addition, the wrist-to-forearm-ratio (WFR) compares the median nerve CSA at the wrist to the CSA 12 cm proximal at the forearm and shows a high sensitivity to detect CTS if the ratio is larger than 1.4 [12].

However, in diabetic patients, still controversies and uncertainties exist as to how far the diagnosis criteria for CTS may be valid for electrodiagnostic testing [13,14] and nerve ultrasound as well [15]. A major factor is generally seen in the coexistence of diabetic neuropathy, which also influences carpal tunnel measurements [16]. Most of the studies evaluated diabetic patients with the clinical diagnosis of CTS. 

The clinical question here is how to interpret electrodiagnostic testing and ultrasound results for suspected CTS in patients with type 2 diabetes. Therefore, the aim of this study is to analyze patients with type 2 diabetes independent from the medical history of CTS or diabetic neuropathy in order to compare CTS symptoms (measured by the Boston Carpal Tunnel Questionnaire), diabetic neuropathy, nerve conduction studies, and peripheral nerve ultrasound of the median nerve.

## 2. Materials and Methods

### 2.1. Patients

We analyzed a database of patients with type 2 diabetes mellitus who participated in the still-ongoing SELECT study (Sonographic and electrophysiological characterization of peripheral nerves in patients with type 2 diabetes, German Clinical Trials Register DRKS00023026). All patients presented in the tertiary care outpatient clinic for diabetology at Jena University Hospital. The data were collected prospectively between September 2020 and April 2022. All participants gave written informed consent. The study was approved by the local ethics committee (number 2019-1416-BO).

Inclusion criteria were patients with type 2 diabetes, age between 40 and 85 years, willing to fill out questionnaires, and willing to undergo nerve conduction studies and peripheral nerve ultrasound. Exclusion criteria were known other etiologies for polyneuropathy (such as alcohol abuse, inflammatory polyneuropathies, etc.), rheumatic disease, peripheral arterial occlusive disease, active malignant tumor disease, and history of chemotherapy and CTS surgery.

### 2.2. Assessments

Several baseline parameters were collected: age, gender, duration of diabetes in years, body mass index (kg/m^2^), HbA_1c_ (mmol/mol), and glomerular filtration rate (mL/min). HbA_1c_ was measured using high-performance liquid chromatography (TOSOH-Glykohaemoglobin-Analyzer HLC-723 GhbV, Tosoh Corporation, Tokyo, Japan).

Symptoms and deficits due to CTS were inquired using the Boston Carpal Tunnel Questionnaire (BCTQ) [17], which consisted of two parts: the Symptom Severity Scale (SSS) and the Functional Status Scale (FSS). The SSS included eleven questions and the answers ranged from 1 (no pain or difficulties) to 5 (severe/permanent pain or difficulties). The score (ranging from 11 to 55) discerns five degrees of severity (0 = asymptomatic to 4 = severely affected). The SSS is performed for each hand separately. The FSS includes eight activities in daily life, which are scored from 1 (no difficulties) to 5 (not feasible). The score (ranging from 8 to 40) also discerns five degrees of severity (0 = asymptomatic to 4 = severely affected). The German version of the BCTQ has been shown to have sufficient internal consistency, reliability, and validity to assess the health status in CTS [18].

Subjective symptoms due to diabetic polyneuropathy were evaluated using the Neuropathy Symptom Score (NSS) and the severity of sensory deficits using the Neuropathy Disability Score (NDS) [19]. NSS asks for sensory symptoms in the legs (burning, numbness, tingling, fatigue, cramping), the localization, time of appearance, and improvements. Scores of 3–4 imply mild, 5–6 moderate, and 7–10 severe symptoms. The NDS checks ankle reflexes, vibration perception threshold (tuning fork), pain sensitivity (pin-prick), and temperature sensitivity. Scores of 3–5 imply mild, 6–8 moderate, and 9–10 severe deficits. Based on NSS and NDS scores, diabetic polyneuropathy can be diagnosed; if the NDS is between 6 and 8 or NDS is between 3 and 5 and NSS is between 5 and 6 [20].

Principally, the right median nerve was measured with nerve conduction studies (NCS) and peripheral nerve ultrasound, except if there was a pathology at the right wrist (such as complex regional pain syndrome, amputation, status after surgery, fractures, and others). In this exception, the left median nerve was measured (in 11 patients). The examiner was blinded with respect to the existence of CTS symptoms.

Nerve conduction studies (NCS) of the median nerve were performed by an experienced neurologist using a Medelec Synergy device (Synergy 15.0; Viasys Healthcare, Natus Europe GmbH, Planegg, Germany). Measurements were carried out on the median nerve (on the same side as the ultrasound measurements). Here, we measured the distal motor latency (DML), the amplitude of compound muscle action potential (CMAP) of the abductor pollicis brevis muscle, the sensory nerve conduction velocity (CV), and the amplitude of the sensory nerve action potential (SNAP) measured at the second finger (Figure 1A–C). The distance over the wrist between stimulation and recording electrode for motor NCS was kept constant at 7 to 8 cm. Skin temperature was controlled to be between 32 and 34 °C. Cut-off values in our laboratory for the median nerve are (according to [21]) DML 4.2 ms, CMAP amplitude 5.0 mV, sensory CV 45 m/s, and SNAP amplitude 6.9 µV.

Sonographic examinations were performed by an experienced neurologist using a high-resolution ultrasound device (Mindray M7, Medical Australia Ltd., Ultrasound systems, Darmstadt, Germany) with a 14 MHz linear-array transducer. The median nerve was measured at the inlet of the carpal tunnel (at the level of the pisiform bone) and 12 cm proximal of the wrist at the forearm. We measured the CSA using direct-tracing technique around the inner margin of the hyperechoic epineural sheath (Figure 1B,D) and calculated the wrist-to-forearm-ratio. For the CSA of the median nerve at the level of the pisiforme bone, a cut-off value of 10 mm^2^ showed good diagnostic utility and a wrist-to-forearm-ratio of ≥1.4 showed a high sensitivity in the general population [22].

### 2.3. Statistics

All data were analyzed with the Statistical Package for the Social Sciences software (SPSS version 25.0; IBM Corporation, Armonk, NY, USA). The values were presented as mean and standard deviation (SD) or as numbers and percentages. First, we described the cohort using descriptive statistics.

Spearman correlations were used to analyze correlations between nerve conduction studies and peripheral nerve ultrasound. Unpaired t-test was used to analyze differences of measurements between patients with and without CTS symptoms and between patients with and without diabetic polyneuropathy. Linear regression was used to evaluate potential influences of clinical parameters on electrodiagnostic measurements and sonographic measurements. Finally, receiver operating characteristic (ROC) analysis was used to evaluate the potential of these measurements to differentiate between patients with and without CTS symptoms. For all analyses, a *p* value < 0.05 was considered statistically significant. 

## 3. Results

### 3.1. Patients

At the time point of analysis, 88 patients were included in the SELECT cohort and were screened for eligibility for this study. Nine patients had CTS surgery before and 10 patients did not answer the BCTQ. Therefore, 19 patients had to be excluded from analysis. Thus, 69 patients with type 2 diabetes (26 female and 43 male) were finally included into this study. Table 1 and Table 2 show the baseline characteristics of the patients. Fifty patients reported no CTS typical symptoms at the analyzed hand (BCTQ SSS = 0) and 19 patients only reported predominantly mild CTS symptoms. In contrast, 49 patients were diagnosed having typical signs of diabetic polyneuropathy (when NDS was between 6 and 8 or NDS was between 3 and 5 and NSS was between 5 and 6, according to [20]).

### 3.2. Nerve Conduction Studies

Distal motor latencies of the right median nerve larger than 4.2 ms were found in 34 patients, and in two patients, no CMAP could be measured. Thirty-four patients showed sensory nerve conduction velocities slower than 45 m/s and five patients had no SNAPs. Figure 2 shows the measurements of nerve conduction studies. As expected, there was a correlation between DML and sensory conduction velocity (Spearman correlation coefficient of −0.532, *p* < 0.001). Using the Bland classification of neurophysiological severity of CTS [23], nine patients showed mild, twenty-two moderate, four severe, and two extremely severe neurophysiological measurements.

DML did not show statistically significant differences between patients with and without CTS symptoms (Figure 2C, *t*-test: T = 1.151, *p* = 0.254) and between patients with and without diabetic polyneuropathy (Figure 2D, *t*-test: T = 1.465, *p* = 0.148). 

### 3.3. Peripheral Nerve Ultrasound

Median nerve CSA at the wrist was between 10 and 12 mm^2^ in 16 patients, between 12 and 15 mm^2^ in 24 patients, and larger than 15 mm^2^ in 21 patients. Sixty-one patients had a WFR ≥ 1.4. Figure 3 shows the ultrasound measurements of the median nerve.

Median nerve CSA at the wrist and wrist-to-forearm-ratio did not show any statistically significant differences between patients with and without CTS symptoms (Figure 3C, CSA: *t*-test, T = 0.621, *p* = 0.537; wrist-to-forearm-ratio: *t*-test, T = 0.161, *p* = 0.873) nor between patients with and without diabetic polyneuropathy (Figure 3D, CSA: *t*-test, T = 1.273, *p* = 0.207; wrist-to-forearm-ratio: *t*-test, T = 0.120, *p* = 0.905).

### 3.4. Interactions

A correlation (Figure 4) between DML and median nerve CSA at the wrist (Spearman correlation coefficient of 0.406, *p* = 0.001) and between DML and wrist-to-forearm-ratio (Spearman correlation coefficient of 0.324, *p* = 0.009) could be found.

Linear regression showed body mass index being a predictive variable for median nerve CSA and HbA_1c_ being a predictive variable for wrist-to-forearm-ratio (Table 3), although both had small R^2^ (0.065 and 0.068, respectively). 

Receiver operating characteristic (ROC) analysis showed that neither distal motor latency, cross-sectional area of the median nerve at the wrist, nor the wrist-to-forearm-ratio were able to distinguish between diabetic patients with and without symptoms for CTS (Figure 5). Area under the curve was 0.632 for the distal motor latency, 0.473 for the median nerve CSA, and 0.546 for the wrist-to-forearm-ratio.

## 4. Discussion

Symptoms characteristic of CTS such as pain, numbness, and/or tingling in the median nerve distribution in the hands have been shown to have a prevalence of 14.4% in the general population, while CTS typical changes in nerve conduction studies of the median nerve have a prevalence of 4.9% [24]. According to the clinical diagnosis of CTS, the prevalence of CTS was 2% in the general population, 14% in diabetic patients without neuropathy, and 30% in patients with diabetic neuropathy [13]. 

In our study, patients with type 2 diabetes were examined regardless of having CTS or not. In this cohort, 50 patients did not report typical symptoms of CTS (according to the BCTQ), and the other 19 patients predominantly complained about mild symptoms. In contrast, 49 patients showed typical clinical signs of diabetic polyneuropathy (due to the NSS and NDS).

Electrodiagnostic testing is the technical gold standard to evaluate CTS [7]. It has been shown that approximately one quarter of diabetic patients had an electrophysiological, but clinically asymptomatic CTS while only 7.7% also had CTS symptoms [25]. A more recent study [14] found about 6.8% of persons with diabetes showing typical electrophysiological signs of CTS being clinically asymptomatic. Thus, it was suggested that asymptomatic CTS constellation in nerve conduction studies in diabetic patients are related to increased vulnerability of peripheral nerves at entrapment sites [14].

In our study, half of the patients showed a pathological increase in DML according to the cut-off values in our lab. Referring to the Bland classification [23] of electrophysiological severity of CTS, 37 patients met the diagnosis criteria for CTS and six patients were classified as severe or extremely severe. Thus, many more patients met the electrophysiological criteria for CTS diagnosis, while considerably fewer patients reported typical symptoms. The major hallmark of our study was that nerve conduction study was not able to differentiate between diabetic patients with CTS symptoms from diabetic patients without.

However, we did not use additional nerve conduction studies to increase sensitivity such as comparison studies of sensory-latency difference between the 2nd and 5th digit, between the median and ulnar part of the 4th digit, or between the median and radial thumb [26], which may be of additional value in diabetic patients.

Considering the results of peripheral nerve ultrasound in our study, the median nerve CSA at the wrist was larger than 12 mm^2^ in 45 patients, and the wrist-to-forearm-ratio was larger than 1.4 in 61 patients.

It has been suggested before that sonographic assessment for the diagnosis of CTS requires a different cut-off value for diabetic patients [27]. A cut-off value for CSA of the median nerve at the wrist to diagnose coexisting CTS and diabetic polyneuropathy of 11.6 mm^2^ was suggested (in contrast to the cut-off used for the diagnosis of CTS in nondiabetic patients of 9.2 mm^2^) [28]. Others suggested a cut-off value of CSA at the wrist for CTS confirmation of more than 13 mm^2^ in both diabetic and nondiabetic patients [29]. Overall, the cut-off values for CSA abnormality in CTS vary considerably in different studies and no consensus exists on a specific optimum cut-off value [9]. 

Nevertheless, ultrasound measurements in our study were not able to differentiate between patients with CTS symptoms and patients without, and even not between patients with polyneuropathy and those without. 

A recent meta-analysis [15] of CSAs of the median nerve at the wrist level described larger CSA measurements in patient groups with both CTS and diabetes than in patients with CTS only and patients with diabetes only, and the smallest CSAs in normal controls. The wrist-to-forearm-ratio in CTS patients with diabetes was significantly lower than in nondiabetic patients, and no difference between the wrist-to-forearm-ratio could be demonstrated between diabetics with and without CTS [30]. It was suggested that an increase in median nerve CSA without change in the wrist-to-forearm-ratio might be an indicator of diabetic polyneuropathy [31]. However, this assumption could not be verified in our study. 

In a study of patients with typical clinical symptoms of CTS, patients with diabetes tended to have a longer latency, smaller amplitude, and lower conduction velocity in nerve conduction studies compared to patients without diabetes mellitus, but the ultrasound CSA values did not differ significantly [32]. In contrast, it was found in a comparison between diabetic patients with symptomatic and asymptomatic CTS that the symptoms of CTS in patients with diabetes are related to CSA of the median nerve [33].

In a small percentage of patients with median nerve entrapment at the carpal tunnel, the CSA is abnormal at the outlet rather than the inlet of the carpal tunnel. However, we measured the CSA at the inlet only. CSA measurements at the outlet had possibly shown other results. In addition, other promising ultrasound techniques were not used in this study—particularly, the evaluation of echogenicity, the intraneural blood flow using Doppler ultrasound [9], or ultrasound elastography—to assess changes in stiffness of the nerve [34,35]. These techniques may provide additional information for CTS diagnosis but need more clinical evaluation.

Polyneuropathy is a common complication in diabetes mellitus [20]. In our study population, 49 (out of 69) patients had polyneuropathic symptoms. However, diabetic polyneuropathy causes alterations in nerve conduction studies [36,37] and in peripheral nerve ultrasound as well [27,38,39]. Generally, diabetic polyneuropathy leads to an enlargement of peripheral nerve CSAs with particular nerve enlargement at entrapment sites [27,40]. These sonographic alterations are explicitly less pronounced than those found in demyelinating polyneuropathies [41,42].

The presence of diabetic polyneuropathy has been shown to be associated with an increase in CSA of the median nerve at the carpal tunnel [16,31]. In contrast, others found median nerve CSA at the wrist significantly smaller in patients with CTS and diabetic polyneuropathy compared with diabetic patients with CTS only [15,43]. 

We found a correlation between distal motor latency and median nerve CSA at the wrist and also between distal motor latency and wrist-to-forearm-ratio, which shows that sonographically enlarged nerves also have slower conduction velocities. This was already shown generally in polyneuropathy [42,44] but also in CTS measurements in patients with diabetic neuropathy [45]. Moreover, it demonstrates that the measurements in our study show the same physiological relationship as those found in other studies and, therefore, seem conclusive. Nevertheless, NCS is generally seen to be superior to ultrasound for the identification of superimposed CTS in diabetic polyneuropathy [45,46].

Besides the existence of polyneuropathy, there may be additional factors able to influence carpal tunnel measurements. In our cohort, we found a small but significant influence of body mass index on median nerve CSA and HbA_1c_ on wrist-to-forearm-ratio. It has been shown that CTS was significantly associated with high body mass index in diabetic patients [47] and generally in the normal population also [48,49]. In addition, higher levels of HbA_1c_ and plasma glucose levels were shown to be associated with an increased risk for CTS in diabetic patients [50]. This shows that the quality of long-term blood glucose control (measured by HbA_1c_) has an impact on CTS ultrasound measurements and, therefore, should be taken into account. It has to be noted that body mass index and HbA_1c_ also have a strong correlation to each other [51,52].

Limitations of the study are the relatively small sample size and the single-center character of the study. All patients were cared for in a tertiary care outpatient clinic for diabetology, which may introduce some selection bias towards patients with potentially more complicated diabetes mellitus. In addition, the study was not primarily designed to study CTS in diabetes. If electrodiagnostic testing and ultrasound had been conducted on the most symptomatic wrist, the results would possibly have been different. However, BCTQ scores were not strikingly different between the left and right hand. Nevertheless, strengths of the study are the use of standardized scores for CTS and diabetic neuropathy in addition to standardized electrophysiological and ultrasound measurements.

## 5. Conclusions

In conclusion, the major finding of our study was that neither the distal motor latency of the median nerve, the cross-sectional area of the median nerve at the wrist, nor the wrist-to-forearm-ratio were able to distinguish between diabetic patients with and without CTS symptoms. This may especially be caused as electrodiagnostic testing and peripheral nerve ultrasound of the carpal tunnel in diabetic patients may significantly be altered due to the existence of additional factors such as diabetic neuropathy, but also body mass index, hyperglycemia, and others. Therefore, it is advisable to primarily rely upon typical symptoms and clinical signs of CTS in diabetic patients. Results of electrodiagnostic testing and peripheral nerve ultrasound have to be interpreted with caution and additional factors have to be considered, especially the existence of diabetic polyneuropathy.

## Figures and Tables

**Figure 1 jcm-11-03374-f001:**
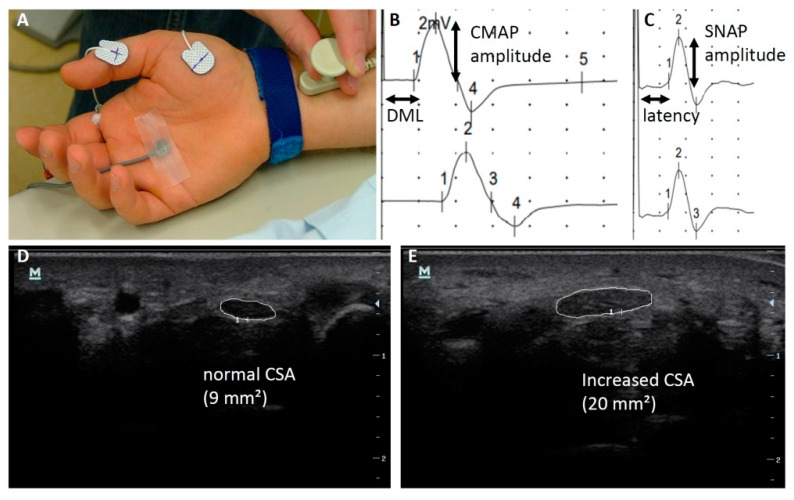
Nerve conduction studies and peripheral nerve ultrasound of the median nerve at the wrist. (**A**) Placement of electrodes. (**B**) Motor nerve conduction study. (**C**) Sensory nerve conduction study. (**D**) Normal cross-sectional area (CSA) of the median nerve. (**E**) Increased CSA of the median nerve. Abbreviations: CMAP, compound motor action potential; CSA, cross-sectional area; DML, distal motor latency; SNAP, sensory nerve action potential.

**Figure 2 jcm-11-03374-f002:**
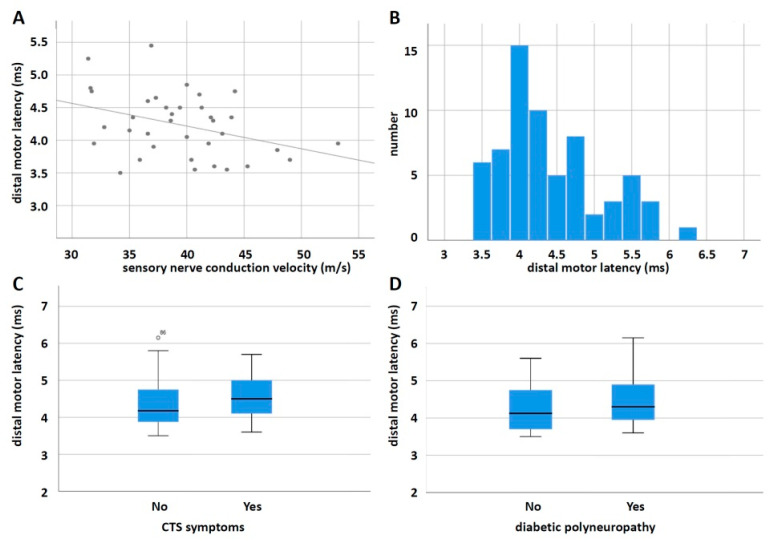
Nerve conduction studies of the median nerve. (**A**) Scatter plot of sensory nerve conduction velocity and distal motor latency. (**B**) Histogram of distal motor latencies. (**C**) Box plots of distal motor latencies in patients with and without CTS symptoms. (**D**) Box plots of distal motor latencies in patients with and without diabetic polyneuropathy.

**Figure 3 jcm-11-03374-f003:**
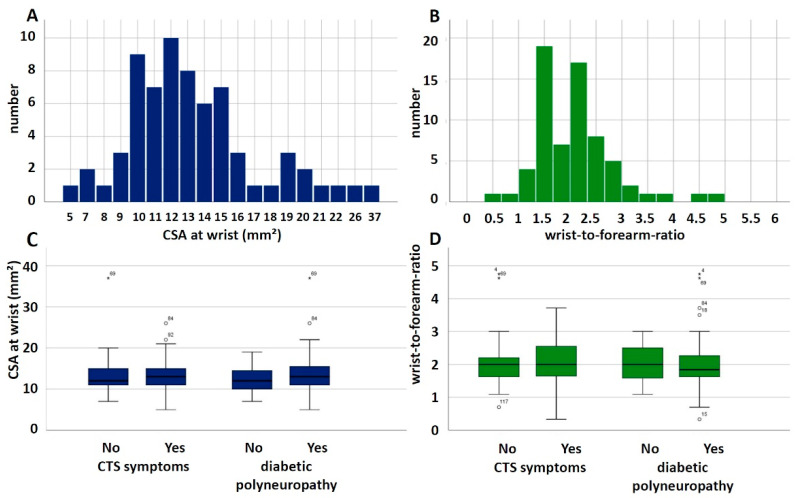
Peripheral nerve ultrasound measurements of the median nerve at the wrist. (**A**) Histogram of the CSA at the wrist. (**B**) Histogram of the wrist-to-forearm-ratio. (**C**) Box plots of median nerve CSA at the wrist in patients with and without CTS symptoms and patients with and without diabetic polyneuropathy. (**D**) Box plots of wrist-to-forearm-ratio in patients with and without CTS symptoms and patients with and without diabetic polyneuropathy.

**Figure 4 jcm-11-03374-f004:**
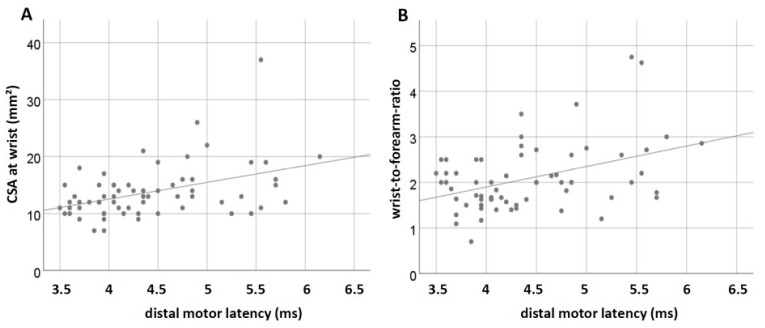
Scatterplots of distal motor latencies and peripheral nerve ultrasound measurements. (**A**) Distal motor latencies and median nerve CSA at the wrist. (**B**) Distal motor latencies and wrist-to-forearm-ratio.

**Figure 5 jcm-11-03374-f005:**
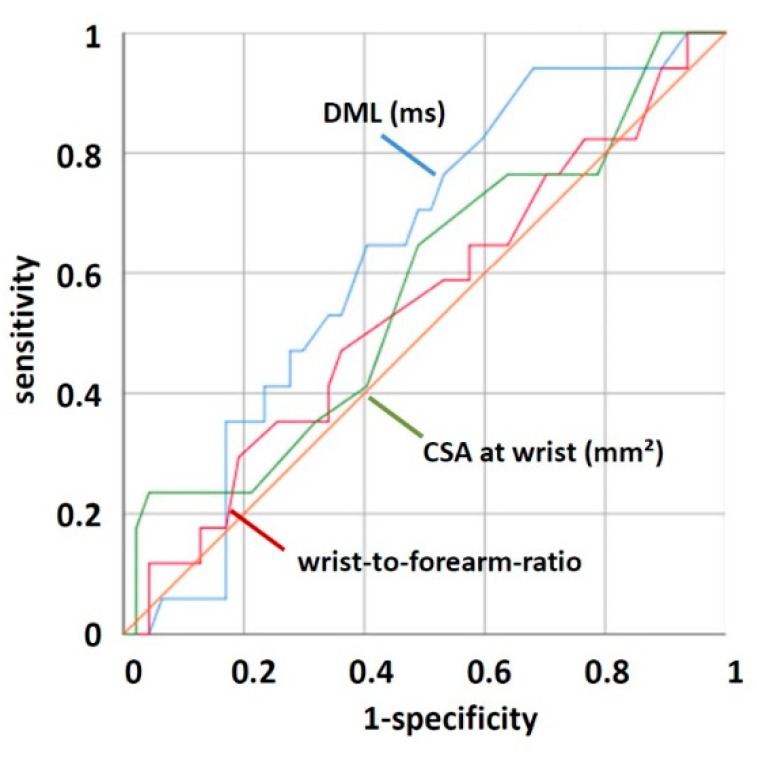
ROC curve analysis shows that DML, median nerve CSA at the wrist, and wrist-to-forearm-ratio were not suited to distinguish between diabetic patients with and without symptoms for CTS.

**Table 1 jcm-11-03374-t001:** Baseline characteristics of the patients (n = 69), categorical variables.

Variable		n	%
Sex	Female	26	37.7
	Male	43	62.3
BCTQ SSS at the measured hand	0	50	72.5
	1	17	24.6
	2	1	1.4
	3	1	1.4
BCTQ SSS right hand	0	50	72.5
	1	17	24.6
	2	1	1.4
	3	1	1.4
BCTQ SSS left hand	0	51	73.9
	1	16	23.2
	2	1	1.4
	3	1	1.4
BCTQ FSS	0	50	72.5
	1	18	26.1
	4	1	1.4
Polyneuropathy if NDS > 5 or (NDS > 2 and NSS > 4)	Yes	49	71.0
	No	20	29.0
NSS	No symptoms (0–2)	27	39.1
	Mild symptoms (3–4)	10	14.5
	Moderate symptoms (5–6)	16	23.2
	Severe symptoms (7–10)	17	24.6
NDS	No deficits (0–2)	9	13.2
	Mild deficits (3–5)	18	26.5
	Moderate deficits (6–8)	26	38.2
	Severe deficits (9–10)	15	22.1
CTS symptoms and diabetic polyneuropathy	Asymptomatic + no neuropathy	18	26.1
	CTS symptoms only	2	0.3
	Diabetic neuropathy only	32	46.4
	CTS symptoms and neuropathy	17	24.6

Abbreviations: BCTQ, Boston Carpal Tunnel Questionnaire; FSS, Functional Status Scale; NDS, Neuropathy Disability Score; NSS, Neuropathy Symptom Score; SSS, Symptom Severity Scale.

**Table 2 jcm-11-03374-t002:** Baseline characteristics of the patients (n = 69), metric variables.

Variable	Mean	SD	Minimum	Maximum
Age (years)	66.77	9.72	44	82
Duration of diabetes (years)	14.72	8.95	0.63	38
Body mass index (kg/m^2^)	32.42	6.17	20.1	48.0
HbA_1c_ (mmol/mol)	59.08	10.94	27.98	82.51
Glomerular filtration rate (mL/min)	71.79	20.64	27.92	107.25
CSA median nerve at wrist (mm^2^)	13.53	4.74	5.00	37.00
CSA median nerve at forearm (mm^2^)	7.03	2.09	4.00	15.00
Wrist-to-forearm-ratio	2.05	0.76	0.33	4.75
Distal motor latency median nerve (ms)	4.42	0.67	3.50	6.15
CMAP amplitude median nerve (mV)	10.12	4.39	0	22.30
Sensory nerve CV median nerve (m/s)	43.26	7.84	29.0	63.6
SNAP amplitude median nerve (µV)	13.31	9.64	0	48.9
NSS (0–10 points)	3.84	3.23	0	9
NDS (0–10 points)	6.10	2.75	0	10

Abbreviations: CMAP, compound motor action potential; CSA, cross-sectional area; CV, conduction velocity; NDS, Neuropathy Disability Score; NSS, Neuropathy Symptom Score; SNAP, sensory nerve action potential.

**Table 3 jcm-11-03374-t003:** Linear regression.

Variable	Coefficient	Standard Error	*p*	Beta
**Model1 with median nerve CSA as dependent variable (R^2^ = 0.065)**				
Constant	7.269	8.375	0.027	
Body mass index	0.197	0.102	0.047	0.255
**Model2 with wrist-to-forearm-ration as dependent variable (R^2^ = 0.068)**				
Constant	1.036	3.206	0.052	
HbA_1c_	0.018	0.097	0.043	0.260

## Data Availability

The data used to support the findings of this study are available from the corresponding author upon request.

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
