# Peer review of "Electrodiagnostic Testing and Nerve Ultrasound of the Carpal Tunnel in Patients with Type 2 Diabetes"

_jcm, 2022, doi:10.3390/jcm11123374_

Round 1
Reviewer 1 Report
Dear authors,
thank you for your manuscript "Electrodiagnostic testing and nerve ultrasound of the carpal tunnel in patients with type 2 diabetes”. It is an interesting article on a very important and potentially growing number of patients.
However, I have comments I think need to be addressed. In general, the manuscript style (abbreviations, English spell check, capital letters, tables, etc) need to be carefully checked. As table 2 did not show as a whole in the manuscript, commenting on discussion was impossible.
Warmly,
your reviewer
Author Response
Dear Editor,
we thank the editor and the reviewers for reviewing the manuscript and sincerely appreciate the opportunity to submit a revised and improved version for publication. We are grateful for the feedback and the extremely constructive comments from the reviewers. We found the comments very encouraging and have edited the manuscript according to their comments.
As requested, please find below the list of our responses to the comments made by the reviewers:
Reviewer 1
COMMENT: thank you for your manuscript "Electrodiagnostic testing and nerve ultrasound of the carpal tunnel in patients with type 2 diabetes”. It is an interesting article on a very important and potentially growing number of patients.
ANSWER: We thank the reviewer for this positive feedback and comment.
COMMENT: However, I have comments I think need to be addressed. In general, the manuscript style (abbreviations, English spell check, capital letters, tables, etc) need to be carefully checked. As table 2 did not show as a whole in the manuscript, commenting on discussion was impossible.
ANSWER: We carefully revised the style of the manuscript (especially see later comments).
In general:
COMMENT: I think the clinical question here is how to interpret ENMG and ultrasound studied for suspected CTS in type 2 diabetic patient. However, if this is the case, it should be clear to readers, now it is not.
ANSWER: We specified this in the introduction: ‘The clinical question here is how to interpret electrodiagnostic testing and ultrasound results for suspected CTS in patients with type 2 diabetes. Therefore, the aim of this study was to….’ (lines 63-65 ).
COMMENT: Using abbreviations must be corrected. In both figures and text, there is heterogeneity in using them, for example CSA appears with no definition in figures and CTS is not used throughout the text.
ANSWER: We changed this accordingly throughout the text.
COMMENT: Using percentages in a patient series of only 69 is not valid.
ANSWER: Now we present the patient numbers throughout the manuscript.
Abstract
COMMENT: lines 20-22: ”19 patients reported predominantly mild CTS symptoms, 71% showed signs of diabetic polyneuropathy. 53.6% met the electrophysiological diagnosis criteria for CTS, 8.7% were classified as severe or extremely severe.” This sentence is a bit confusing. I understand that 19/69 patients had mild CTS symptoms, but to whom do the percentages presented next refer to? Those 19 patients or all patients?
ANSWER: Now we present the patient numbers to make it clear. (lines 20-24)
Introduction
COMMENT: line 35: ”A significant risk factor” – is there such thing as insignificant risk factor? I recommend editing this sentence, for example to only as you state next ” it has been shown that the incidence of carpal tunnel syndrome is increased in diabetic patients”. A factor that increases the incidence in a certain population is a risk factor. Also, as you represent abbreviation for carpal tunnel syndrome (CTS) you should stick to this throughout the whole text.
ANSWER: We changed it as follows: ‘Risk factors for CTS are diabetes mellitus, obesity, metabolic syndrome, thyroid dysfunction, or rheumatic diseases and others [2,3]. It has been shown that the incidence of CTS is increased in diabetic patients [4].’ (lines 36-39)
COMMENT: the last paragraph, lines 62-66: This paragraph should define the study hypothesis to keep the reader interested. I’d like to edit this to something like ”The aim of this study was to… in patients with type 2 diabetes mellitus.”
ANSWER: We changed it as follows: ‘Therefore, the aim of this study was to analyze patients with type 2 diabetes….’ (lines 64-68)
Materials & methods
COMMENT: Why was the ENMG study ran for the right wrist and not on the symptomatic wrist?
ANSWER: The study protocol included the examination of one median nerve only. ’The examiner was blinded with respect to the existence of CTS symptoms.’ We included this statement in the material section (lines 112-113). We think that this is an essential characteristic of this study.
Results
COMMENT: lines 153-154: ”In total, 69 patients with type 2 diabetes (26 female and 43 male) were included into the study.” How many were excluded as these were the ones meeting the inclusion criteria?
ANSWER: We specified it as follows: ‘At the time point of analysis 88 patients were included in the SELECT cohort and were screened for eligibility for this study. Nine patients had CTS surgery before and 10 patients did not answer the BCTQ. Therefore, 19 patients had to be excluded from analysis. Thus, 69 patients with type 2 diabetes (26 female and 43 male) were finally included into this study.’ (lines 157-161)
COMMENT: As patients were only 69, using percentages is not valid.
ANSWER: We changed it into numbers (throughout the manuscript).
Table 1:
COMMENT: maximum lacks capital letter in the beginning
ANSWER: We corrected it (line 170).
COMMENT: ENMG and ultrasound were performed ipsilaterally, but BCTQ is presented for right and left side. Would it be relevant to present them as measured side and nonmeasured side?
ANSWER: This is correct. We included the BCTQ information according to the measured side in the table ( table 1). The calculation of statistics concerning these parameters was not altered as we already calculated it for the measured side before.
Reviewer 2 Report
1. Nerve conduction studies: It appears that only median nerve studies were done. Most labs do comparison studies ( median:ulnar motor 2nd Lumb/Interosseus, median:ulnar sensory 4th digit, and median: superficial radial thumb) to increase sensitivity; this is particularly relevant in diabetic patients. Wonder if the conclusions may have been different with the use of these measurements.
2. Ultrasound: In a small % of patients with median nerve entrapment at the carpal tunnel, the CSA is abnormal at the outlet rather than the inlet of the carpal tunnel. Wonder if the results may have been different if the maximum CSA (inlet or outlet) was used instead of confining to the inlet CSA.
Author Response
Dear Editor,
we thank the editor and the reviewers for reviewing the manuscript and sincerely appreciate the opportunity to submit a revised and improved version for publication. We are grateful for the feedback and the extremely constructive comments from the reviewers. We found the comments very encouraging and have edited the manuscript according to their comments.
As requested, please find below the list of our responses to the comments made by the reviewers:
Reviewer 2
COMMENT: 1. Nerve conduction studies: It appears that only median nerve studies were done. Most labs do comparison studies (median:ulnar motor 2nd Lumb/Interosseus, median:ulnar sensory 4th digit, and median: superficial radial thumb) to increase sensitivity; this is particularly relevant in diabetic patients. Wonder if the conclusions may have been different with the use of these measurements.
ANSWER: This is correct. We therefore point it out in the discussion now: ‘However, we did not use additional nerve conduction studies to increase sensitivity such as comparison studies of sensory-latency difference between 2nd and 5th digit, between the median and ulnar part of the 4th digit, or between the median and radial thumb [26], which may be of additional value in diabetic patients.’ (lines 265-268)
COMMENT: 2. Ultrasound: In a small % of patients with median nerve entrapment at the carpal tunnel, the CSA is abnormal at the outlet rather than the inlet of the carpal tunnel. Wonder if the results may have been different if the maximum CSA (inlet or outlet) was used instead of confining to the inlet CSA.
ANSWER: Yes, we address this as follows now: ‘In a small percentage of patients with median nerve entrapment at the carpal tunnel, the CSA is abnormal at the outlet rather than the inlet of the carpal tunnel. However, we measured the CSA at the inlet only. CSA measurements at the outlet had possibly shown other results.’ (lines 298-301)
Yours sincerely,
Bianka Heiling
Round 2
Reviewer 1 Report
Dear autrhors,
thank you for the revised manuscript! It was improved, and I have no further comments.
Warmly,
your reviewer